# In the Era of mRNA Vaccines, Is There Any Hope for HIV Functional Cure?

**DOI:** 10.3390/v13030501

**Published:** 2021-03-18

**Authors:** Ignasi Esteban, Carmen Pastor-Quiñones, Lorena Usero, Montserrat Plana, Felipe García, Lorna Leal

**Affiliations:** 1Institut d’Investigacions Biomèdiques August Pi i Sunyer (IDIBAPS), 08036 Barcelona, Spain; iesteban@clinic.cat (I.E.); capastor@clinic.cat (C.P.-Q.); usero@clinic.cat (L.U.); mplana@clinic.cat (M.P.); fgarcia@clinic.cat (F.G.); 2Infectious Diseases Department, Hospital Clínic, University of Barcelona, 08036 Barcelona, Spain

**Keywords:** mRNA, vaccines, HIV, infectious diseases

## Abstract

Over 36 million people worldwide are infected with HIV. Antiretroviral therapy (ART) has proven to be highly effective to prevent HIV-1 transmission, clinical progression and death. Despite this success, the number of HIV-1 infected individuals continues increasing and ART should be taken for life. Therefore, there are two main priorities: the development of preventive vaccines to protect from HIV acquisition and achieve an efficient control of HIV infection in the absence of ART (functional cure). In this sense, in the last few years, there has been a broad interest in new and innovative approaches such as mRNA-based vaccines. RNA-based immunogens represent a promising alternative to conventional vaccines because of their high potency, capacity for rapid development and potential for low-cost manufacture and safe administration. Some mRNA-based vaccines platforms against infectious diseases have demonstrated encouraging results in animal models and humans. However, their application is still limited because the instability and inefficient in vivo delivery of mRNA. Immunogens, design, immunogenicity, chemical modifications on the molecule or the vaccine delivery methods are all crucial interventions for improvement. In this review we, will present the current knowledge and challenges in this research field. mRNA vaccines hold great promises as part of a combined strategy, for achieving HIV functional cure.

## 1. Introduction

Nowadays, vaccines can prevent millions of transmissible infections and save as many lives around the world. Vaccination is one of the most effective public health interventions to control infectious diseases. HIV-1 has infected more than 36 million people globally, most of them living in developing countries. At the end of 2019, 38 million people were living with HIV, and 32.7 million people have died of AIDS related diseases. Antiretroviral therapy (ART) has significantly reduced morbidity and mortality of people living with HIV (PLHIV). Despite its effectiveness, ART is a life-long treatment and, it is unable to cure or eradicate the infection [1]. Thus, ART can only suppress HIV productive replication in infected cells and has no impact in latent reservoir.

An alternative to life-long ART could be a functional cure and therapeutic vaccines a very promising approach. To avoid life-long use of ART, there have been very interesting proposals for therapeutic vaccines but not very successful in controlling HIV viral load.

Classic vaccine platforms, such as inactivated virus or peptide-based vaccines have had tremendous success in the control and eradication of several human infectious diseases (smallpox, polio, measles, etc.); however, the administration of these vaccines could promote certain complications, such as the reversion to a virulent form in the case of organism-based vaccines [2] or a low CD8 T-cell immune response [3,4], which is essential for clearing out HIV infected cells. New approaches have recently been used based on recombinant viral vectors or autologous dendritic cells (DCs) that induce an HIV-specific immune response but had a limited effect on HIV viral dynamics.

Type and selection of the immunogen are key factors in the design of a vaccine. Thus, in this field, making a gene construct coding for the antigen of interest, instead of a pathogen or a recombinant protein-based vaccine is easier, faster, and avoids the potential risks of working with live pathogens. These advantages are crucial to produce vaccines in the case of an epidemic. Nucleic acid (DNA/RNA) technology has emerged as an alternative to conventional vaccines. In the 1990s, it was published that “naked DNA” injected directly into mice’s muscle resulted in the expression of protein [5]. Several experimental DNA-based vaccines in cancer, autoimmune, allergic, and infectious diseases (HIV, malaria, influenza, and hepatitis B) have been developed since then. Currently, different combinations of DNA and viral vectors are in the preclinical stage and have showed an improvement in the magnitudes of HIV-1-specific CD8^+^, CD4^+^, and T follicular helper (Tfh) cells [6,7]. Furthermore, DNA-based vaccines against HIV infection have also been used in clinical trials [8,9,10] with different outcomes and even though DNA vaccines have a great potential, disappointing results have been obtained mainly because, when used alone, they are poorly immunogenic. Such immunogenicity could be improved with a prime-boost and/or injected with a device, but it will make it expensive and complex to implement clinically.

As with DNA, the administration of RNA vaccines can produce protein expression in vivo [5]. Over the past decade, the advances in research and new technologies have enabled mRNA to become a promising therapeutic tool. Using RNA-based immunogens has some benefits over proteins, dead and live attenuated viruses, as well as DNA-based vaccines as represented in Table 1.

Some studies have indicated that nucleoside modification was an effective approach to enhance mRNA stability and translational capacity but reducing its immunogenicity in vivo [11,12]. After solving the instability problem, in 2008, the first clinical trial assessing a RNA-based vaccine for melanoma was approved [13], and in the next year, an HIV messenger RNA (mRNA)-based vaccine clinical trial was conducted [14]. In an attempt to improve mRNA expression, a synthetic lipid nanoparticle formulation of self-amplifying RNA (LNP/RNA) was tested for the first time. The results showed an increase in antigen production and immunogenicity in vivo, without the need for a viral delivery system [15]. At present, very few phase I and II clinical trials with mRNA vaccines against HIV have been conducted with encouraging results (Figure 1).

This review discusses the advantages of using mRNA as therapeutic vaccine against HIV infection, offering a new and optimistic perspective.

## 2. Types and Molecular Biology of mRNA Vaccines

During the last decade, significant technological innovation and the urgent need to develop more versatile vaccination platforms have made mRNA a promising tool for vaccine development [21]. Currently, there are two types of synthetic mRNA vaccines: conventional or non-replicating (NRM) and self-amplifying (SAM). Conventional mRNA-based vaccines include an opening frame (ORF) encoding the target antigen, 5′ and 3′ untranslated regions (UTRs), and a terminal poly(A) tail (Figure 2) [22]. In SAM, the ORF, also contains all the replicative components derived from positive-stranded mRNA viruses to direct intracellular mRNA self-amplification and abundant protein expression [23,24,25]. The self-amplifications come from the replication machinery, whereas the structural protein sequences are replaced with the gene of interest and the resulting genomes are called replicons.

In general terms, the development of a mRNA vaccine is as follows. Once the antigen of choice is identified or selected by sequencing and bioinformatic approaches, the gene is synthesized and cloned into a DNA template plasmid (pDNA). The mRNA is then transcribed in vitro by using a phage RNA polymerase. When the vaccine is administered to the subject, the mRNA uses the host cell machinery for in vivo translation. This entire process mimics a viral infectious reaction, producing the target antigen and initiating an adaptive immune response. [26].

The 5′ cap structure (m7Gp3N (N: any nucleotide)), the 5′ untranslated region (5′ UTR); the open reading frame (ORF), the 3′ untranslated region (3′ UTR) and the tail poly-A tail are elements of the NRM and SAM vaccines that control protein synthesis by influencing the stability and the interaction with the translation machinery [24,25] (Figure 2).

The presence of a 5′ cap structure is essential for mRNA stability and gene expression. It allows an efficient translation in vivo while protecting mRNA from intracellular nuclease digestion [28].

The 5′ and 3′ UTR regions are necessary for increasing gene expression. These structures may regulate cap-dependent translation initiation (through helicase-mediated remodeling of RNA structures and RNA interactions), cap-independent translation initiation, mRNA adjustments, and other specific translation pathways. Characteristics as the length of the 3′ UTR and 5′ UTR regions and the regulatory elements in both UTRs impact the mRNA stability and expression [29,30]. The poly(A) length is also essential for effective translation and suitable protection [31,32].

Nucleoside base modification and codon optimization also modulate translation efficiency. The content of guanine and cytosine (GC) [33] and the use of modified nucleosides, such as pseudouridine or N-1-methylpseudouridine, are both factors that magnify antigen expression [34]. In studies using unmodified mRNA, the optimization of the coding sequence and mRNA purification has been linked to improved protein production, decreased unwanted inflammatory responses, and enhanced adaptive immune responses [27,35]. However, extensive studies comparing the properties and clinical outcomes of unmodified versus modified nucleosides are still missing.

Purification of mRNA is an important step because contaminants can activate non-specific innate sensors. Some studies have demonstrated that mRNA purification from double-stranded RNA (dsRNA) contaminants can enhance in vivo translation and reduce innate immune activation. This step is vital for RNA-based gene therapy and CAR T cell therapy applications [36,37].

## 3. mRNA Vaccine Delivery Methods

The basic principle behind mRNA vaccination is introducing an exogenous mRNA inside an antigen-presenting cell (APC). In the cytoplasm, the exogenous mRNA will be translated into a protein and later processed and presented as an antigen through HLA to T-cells, initiating an adaptive immune response [22]. The mRNA’s product could also be secreted outside the cell, being used as a source of antigen for antibody production [38,39], and being internalized by APCs to be presented to CD4^+^ T cells and cross-presented to CD8^+^ T cells [40,41].

Besides providing antigenic stimulation, an RNA vaccine also needs to provide an adjuvant activity to achieve an effective immune response. The biochemical structure of RNA has a non-specific immunostimulatory activity by itself, as it can be recognized by multiple innate receptors [42]. TLR-7 and TLR-8 can recognize the ssRNA from NRM vaccines in the endosomes. The dsRNA from SAM vaccines and from contaminants in NRM vaccines can be recognized by RIG-1 and MDA-5 in the cytosol, and by TLR-3 in the endosomes. Additionally it has been described that RAGE could bind RNA on the cell surface [43]. Activation of the innate immune response through this set of receptors could provide the adjuvant activity required to prime the adaptive immune response [42].

One of the crucial points for developing effective mRNA vaccines is delivering the mRNA into the cytoplasm of APCs. Since many years ago, it is well known the feasibility of injecting naked mRNA to produce the coded protein expression in vivo [5]; however, the injection of naked or free mRNA is not very efficient for the generation of an immune response, as it has been shown by many reports [16,44]. Martinon and colleagues [16] demonstrated that in vivo injection of an mRNA encoding for the influenza virus nucleoprotein (NP) could produce a cytotoxic T lymphocytes (CTL) mediated response, but only when it was administered formulated with liposomes. The authors hypothesized that liposomes might protect and improve mRNA uptake by APCs. The response was also dependent on the route of administration. They could only detect a CTL response when the vaccine was administered intravenously or subcutaneously, but not when administered intraperitoneally.

There are some significant drawbacks that limit mRNA vaccines efficiency. One is that in the extracellular medium, free mRNA is easily degraded by nucleases [45]. Another is that passive internalization of free mRNA by cells is not very efficient due to the negative charge of RNA. Finally, after entering the cell, exogenous naked mRNA can easily be degraded in the endo-lysosomal compartments. Several strategies have been used to improve mRNA vaccine delivery and the basic principle is to protect mRNA from degradation while improving APC’s uptake. In the following paragraphs, we will discuss the most popular methods that have been used for in vivo mRNA vaccine delivery and give examples in the context of HIV.

### 3.1. Naked mRNA

As mentioned above, injection of unformulated mRNA is not a very efficient method for inducing an immune response compared to other strategies [16,44]. Often the injection of free or naked mRNA does not produce any response, and when there is one, much higher doses of mRNA are needed compared to other methods [46]. Despite its drawbacks, some promising results have been achieved using naked mRNA for vaccination.

The route of vaccine injection is a key point that greatly impacts mRNA up-taking and trafficking, affecting the elicited vaccine-specific response [38]. A strategy that had been used to improve the immunogenicity of naked mRNA vaccines is intranodal injection. This improvement has been attributed to a more efficient uptake of mRNA by dendritic cells (DCs) [47]. In lymph node (LN), mRNA is mostly internalized just by DCs, whereas in the periphery, many other cells can retain mRNA [42] lowering the dose available for presentation. Moreover, in LN, DCs can present their content without moving from the periphery to the secondary lymphoid organs, making the whole process more efficient.

In the context of HIV, intranodal injection of naked mRNA has been tested with interesting results [48]. Guardo et al. demonstrated that mice injected intranodally with a naked mRNA encoding a sequence codifying 16 defined HIV CD8^+^ and CD4^+^ T cell epitopes directed against conserved regions of the virus, alone or in combination with an adjuvant, developed a strong in vivo CTL response against the antigens of the vaccine. Using human samples of LN, they also confirmed that naked mRNA could induce maturation of human DCs residents in the LN.

### 3.2. Dendritic Cell Vaccination

Dendritic cells are the most efficient APCs in priming naïve T-cells. After up-taking the antigen, they process it and present it through HLAI and HLAII to CD8^+^ and CD4^+^ T cells, respectively. More than two decades ago, many studies have used DCs as a vehicle for vaccine delivery [17,39]. DCs are charged with the antigen of interest and re-infused to the host as a vaccine [49].

The antigen can be delivered to DCs as peptides, protein, DNA, or viral vectors [39]. Nevertheless, all those methods have some limitations that can be solved using mRNA as an antigen source. There is no need to match HLA with mRNA, every subject process and present its own peptides. RNA is translated into protein in the cytoplasm, so it can be easily processed and presented through the classical HLAI route, allowing for more potent activation of CD8^+^ T cells. Apart, mRNA could be designed to contain specific sequences to direct the content to the HLAII route, therefore also inducing a strong CD4^+^ T cell response. RNA needs to reach the cytoplasm, and since it is a non-integrating molecule, there is no risk of host genome integration as with DNA. Viral vectors are more complex to work with, and they could be potential infectious agents, so security regulations to work with them are stricter. All those characteristics have made RNA an ideal molecule for dendritic cell vaccination.

The most popular and efficient way to introduce exogenous mRNA into DCs is by electroporation [50,51,52]. This method consists in applying an electrical field to the cells to increase the permeability of the cell membrane, allowing the entrance of the mRNA into the cytoplasm. Another popular method for DC transfection is lipofection [39,53]. Lipofection uses tiny lipid vesicles, called liposomes, with a similar biochemical composition as the cell membrane. Liposomes form a complex with mRNA that can be endocytosed or directly fused with the cell membrane, in both cases releasing the mRNA into the cytoplasm of the DC.

RNA transfected DCs had been used in preclinical studies, both in vitro [39] and in vivo [54]. In this report [39], authors demonstrated for the first time that a mRNA-DC vaccine could induce a primary T cell response in vitro against HIV and also they found p24 protein in the supernatant, suggesting that the vaccine may also be capable of inducing a humoral response.

RNA transfected DCs have also been tested in animal models. In this example [54], mRNA transfected CD34-DCs were used to vaccinate macaques against HIV gag protein. In this study, a specific subset of DCs, derived from CD34^+^ hematopoietic stem cells, was used due to its shared properties with Langerhans cells. Vaccination resulted in the generation of poly-specific CD4^+^ and CD8^+^ T cells against gag antigens. The CD8^+^ T cell response against gag antigens increased with every subsequent dose of the vaccine, opening the door at using CD34^+^ DCs as vehicles for therapeutic HIV vaccines.

### 3.3. Formulated mRNA

The formulation of mRNA with cationic carriers represents an alternative way to DCs for in vivo delivery of mRNA. Several formulations of mRNA with biochemical or chemical carriers have been tested for vaccination. The chemical nature of those carriers is diverse, lipidic [44,55], polymeric [56], peptidic [57], or a combination of them [58]. Usually, those carriers are positively charged (cationic) to bind with the negative charges of mRNA.

Compared with naked mRNA, carrier formulated mRNA can resist degradation by extracellular nucleases while improving APCs’ uptake. In the endosomal compartment, the carrier can also facilitate the mRNA’s escape into the cytosol. Compared with DC vaccination, nanoparticle formulated mRNA works similarly for in vivo mRNA delivery [16], but with the advantage that carriers are much easier to produce and to administrate, reducing the cost of a potential treatment considerably.

Cationic lipid formulations are the most used carriers for mRNA vaccination. The first cationic lipid formulations used for mRNA vaccination were liposomes [58]. While liposomes represent a promising vehicle to deliver exogenous mRNA into the cytosol of APCs, they still have some drawbacks that limit their efficacy, low stability, low efficiency of transfection, and high toxicity [59].

An improved version of liposomes is lipid nanoparticles (LNPs). LNPs are a class of particles with different lipid compositions and ratios as well as different sizes and structures formed by different methods [60,61]. LNPs are typically composed of an ionizable lipid, cholesterol, PEGylated lipid, and a helper lipid such as di-stearoyl-phosphatidylcholine (DSPC) [62]. A key component of LNPs that seems to account for the better effectivity is the ionizable lipid. At a physiological pH of 7,4, the ionizable lipid exhibits a minimal positive charge [63], which is thought to improve uptake and reduce its toxic effects. In this work [57,64,65], the authors used a nucleoside modified mRNA, encoding Env-gp160, formulated with an LNP to vaccinate rabbits and macaques against HIV. The vaccine-induced high levels of specific antibodies against gp120 but neutralizing activity started to decline in just 4–6 weeks after immunization. This study demonstrated for the first time that LNPs could be used as an effective vehicle for mRNA vaccination against HIV.

A wide range of cationic polymers have also been tested for mRNA vaccination, with relevant preclinical results [66]. Cationic cyclodextrin-polyethyleneimine 2k conjugate (CP 2k) is a polymer that has been used as a vehicle for intranasal mRNA vaccination against HIV gp120 producing a specific humoral and cellular immune response [44]. On the other hand, when the same mRNA was injected naked, the response was barely detectable, highlighting the carrier’s importance for the vaccine’s effectiveness. Moyo et al. [67] had also used a synthetic polymer for HIV mRNA vaccination in mice. In that case, a commercial transfection reagent based on polyethyleneimine (Polyplus Transfection, Illkirch, France) was used. Using a self-replicative mRNA, encoding the tHIVconsv1 and tHIVconsv2 immunogens (T-cell immunogens designed to target conserved regions of HIV), the authors were able to induce a polyfunctional CD8^+^ and CD4^+^ T cell response that lasted for at least 22 weeks after vaccination. In this study vaccination with naked mRNA induced a relatively strong response, although not as strong as with formulation. This effect could be explained because a self-replicative mRNA was used. All the referred results evidence that formulated vaccines induce better immune responses than naked mRNA.

## 4. Therapeutic mRNA Vaccines in Clinical Trials: The Experience with HIV

Significant efforts have been made in the last 30 years to find a successful vaccine that could functionally cure HIV [68]. We consider an HIV cure as the intervention that leads to HIV remission, suppressing viremia and maintaining viral control in the absence of antiretroviral treatment [69].

Different approaches for therapeutic vaccines have been studied, such as the “classical” whole inactivated virus [70] or a recombinant protein [71] as well as those based on peptides [72], DNA vectors [73] or autologous dendritic cells [74]. Although most immunogens have induced HIV-specific immune responses, they have shown limited efficacy to control viral replication in clinical trials [75]. New approaches based on more innovative immunogens have been proposed, such as mRNA vaccines. To date, only a few HIV mRNA vaccines clinical trials have been conducted (Table 2), most of them using transfected DCs [14,18,76,77,78,79] and another couple evaluating naked mRNA [80,81].

### 4.1. mRNA DC Vaccination

A pilot study evaluating an immunotherapy based on autologous DCs electroporated with RNA encoding CD40L and Gag, Vpr, Rev, and Nef (AGS-004) was conducted by J.P. Routy and collaborators [14] in Canada. Nine chronically HIV-infected men under stable ART received 4 intradermal (ID) injections every 4 weeks for 4 treatments in the axillary lymph node area. During the study, there were only grade 1–2 related adverse events, and half of the subjects had increases in CD8^+^ T cell proliferative responses to the product, which was the primary endpoint. They concluded that the intervention was safe, feasible, and may be associated with CD8^+^ T cell proliferative responses. Later on, a phase 2B, multicenter, 2:1 randomized, double-blind, placebo controlled study was conducted [76]. A total of 35 participants completed treatment and 4 weeks after last vaccination interrupted ART (ATI) during 12 weeks. There was a robust expansion of HIV-specific CTL responses in subjects receiving AGS-004 but had no impact in viremia control since HIV viral load (VL) rebounded in all participants with no significant differences between arms. Moreover, there were no significant differences in the integrated DNA changes. There was no measurable antiviral effect of AGS-004. This treatment was also administrated monthly to 6 suppressed individuals who started ART during acute HIV infection while participating in a single-arm sub-study [77]. All participants had an increased HIV-1 specific CD8^+^ T cell response post-vaccination and underwent ATI. Even though viral rebound occurred in all participants at a median of 29 days, greater expansion of CD8^+^ T cell responses correlated with longer time to rebound (TtR).

Investigators proposed that adding an immunoregulatory molecule could enhance the activity of the product, and very recently, they conducted a study investigating the impact of vorinostat (VOR), a latency reverse agent, combined with AGS-004 on persistent HIV infection [78]. Five HIV-infected ART-treated participants received eight doses of AGS-004 and 20 doses of VOR over approximately 10 months. The intervention was safe and well-tolerated; unfortunately, it had no measurable impact on the replication competent reservoir. Even though one of the participants started ART during acute infection, there was not an increased response compared to those starting ART in a chronic phase.

Ellen Van Gulck and cols [18] electroporated DCs with mRNA encoding HIV-1 subtype B consensus Gag or chimeric Tat-Rev-Nef protein from 6 chronically HIV subtype B-infected individuals under stable ART. Primary endpoint was safety and feasibility of this type of vaccination. The vaccines were safe with mild reactions, and there was an overall increase in magnitude of the IFN-γ response and significant to Gag. They found an overall increased ex vivo virus-inhibiting capacity after vaccination in most of the patients.

A randomized, placebo-controlled trial [79] that enrolled 15 participants under stable ART, investigated if transfected DCs with mRNA encoding HIV-1 Gag and Nef could be able to elicit T cells response in vivo. Randomization was 2:1, and all the participants received 4 ID injections. All participants also received a second ID of autologous DCs pulsed with the neo-antigen keyhole limpet hemocyanin (KLH). Vaccinations were well tolerated, and Nef CD4^+^ T cell proliferative responses were transiently increased in vaccine recipients as compared to placebo. Moreover, participants developed de novo CD4^+^ and CD8^+^ T cell proliferative response to KLH. However, they did not detect significant vaccine-induced boosting of T cell responses.

### 4.2. Naked mRNA Vaccination

In 2013, our research group and in collaboration with other 4 centers in Spain, Belgium, and The Netherlands formed the iHIVARNA consortium to study an mRNA-based therapeutic HIV-1 vaccine. This vaccine consisted of TriMix, a compound mRNA formula encoding for CD40L, CD70 and a constitutively active Toll-like receptor (caTLR4), and HIVACAT T-cell immunogen (HTI), containing 16 fragments of HIV-1 Gag, Pol, Vif, and Nef proteins [48]. A dose escalation, single center, phase I clinical trial was conducted and included twenty-one HIV-1 chronic infected individuals under stable ART. Participants received 3 inguinal intranodal doses of mRNA ultrasound-guided injections following a dose escalation scheme [81]. The intranodal injection of the studied vaccine was feasible, safe, and well-tolerated even with the highest dose; therefore, this dose was selected for a phase II clinical trial. Moreover, vaccine was able to induce moderate HIV-specific immune responses and transiently increased cell-associated HIV-RNA (ca-RNA) expression and ultrasensitive viremia. Phase II was a 3-arm randomized, multicenter, international, double-blind trial comparing the selected vaccine, TriMix alone and placebo [80]. Thirty-two HIV-1 chronic infected individuals on ART were included, completed vaccination, and underwent ATI. Authors concluded that the interventions were safe and well-tolerated. All participants had a viral rebound after a median of 2 weeks, and TtR was not significantly different amongst groups. There was no significant increase in specific T cells responses in the vaccine group compared to placebo, dictating to halt further inclusion of patients for futility. Overall, the studied vaccine did not have a significant effect on viral reservoir kinetics at any time point compared to control groups. Right after ending phase II clinical trial authors were informed that the study product had an error; the RNA sequence contained by mistake a second start codon in front of the HTI immunogen coding sequence [82]. This error was likely to influence HTI expression, but how this could have affected the results remains unclear.

Following search in clinicaltrials.gov (accessed on 11 February 2021), at the moment of writing this review, we found 27 different proposed strategies and candidates. There are 19 vaccine clinical trials ongoing (9 prophylactic vaccines and 5 therapeutic vaccines) and 8 vaccine clinical trials not yet recruiting (5 prophylactic vaccines and 3 therapeutic vaccines). Only one of these, hopefully soon to be performed, clinical trials is planning to evaluate a personalized mRNA-based immunogen and in combination with a broadly neutralizing antibody and a latency reverse agent to achieve a functional cure of HIV infection (H2020 Grant Agreement Number: 731626).

## 5. Improving mRNA Vaccines for HIV

Given the promising expectations and advantages of mRNA-based vaccines, the development of new immunogens and new strategies capable of addressing the limitations encountered so far is a priority. Although mRNA vaccines indeed hold great promises, it is necessary to know more about their action mechanisms to improve them.

As has already been mentioned in this review, conventional mRNAs technology has been investigated in preclinical and clinical studies with siRNA and mRNA structure modifying methods raising importance over the past few years. Revising the 5′ cap structure; controlling the length of the poly(A) tail; including modified nucleotides, codon or sequence optimization; and modulating the 5′ and 3′ UTRs are just some of the items under analysis in optimization of mRNA translation [83].

However, some studies have opted specifically for crucial features as nucleoside-modified and purified mRNA-lipid nanoparticle (mRNA-LNP) [84] due to a therapeutic relevance reinforced by advantages as the generation of potent degradation defense and long-lived antibody responses [21], and ability to efficiently activate T follicular helper (Tfh) cells [85]. In recent studies, the characterization of HIV-1 nucleoside-modified mRNA vaccines in rabbits and rhesus macaques has had promising results, generating antibodies and antibody-dependent cellular cytotoxicity [63]. In conjunction, these results are upholding the carry-on development of nucleoside-modified and purified mRNA-LNP vaccines for HIV.

It is important to find the correct balance between immunostimulatory activity and protein production, as the overstimulation of the innate receptors by RNA could potentially induce its degradation. In this context, optimizing the RNA production processes to eliminate all the dsRNA has been related to improved vaccine effectiveness [86].

The nature of RNA also allows to include adjuvants into the same molecule [48,81]. Finding new combinations of adjuvant molecules that can be included in the RNA sequence or in the vaccine formulation [64,87] is a promising field of research that could improve vaccine’s effectiveness.

As demonstrated by the recent success of SARS-CoV-2 mRNA based-vaccines [88,89], LNPs are ideal delivery systems for mRNA vaccines. To date, in HIV infection, there has not been any clinical trial using nano-encapsulated mRNA-based vaccines. Therefore, one next logical step would be to test co-formulated mRNA in clinical trials. Selecting the right nano-formulation for a vaccine is a tricky challenge, as hundreds of different compounds have been described, and usually, the ones that work well for one mRNA may not work for another. To add more complexity, some of the best working compounds have been patented by companies, so their use may not be available for any new vaccine. A key point for obtaining an effective mRNA vaccine against HIV infection is the immunogen’s design. The immunogen needs to be capable of inducing broad and strong CTL responses; therefore, it has to include epitopes covering the most frequent HLA molecules in the population while at the same time targeting viral proteins in conservative regions to limit immune escape.

## 6. mRNA-Based HIV Functional Cure Strategies

The major challenge for an HIV functional cure is the existence of a latent reservoir [90]. The latent reservoir is a series of cell types (CD4^+^ T cells and myeloid cells) with a competent HIV genome integrated on their DNA, in a latent state of non-transcription. The reservoir is established in the first stages of infection and can persist even when ART has started early. This state of repressed transcriptional activity makes the virus undetectable to the immune system. As viral proteins are not traduced, there is no antigen presentation through MHCI, and therefore, CD8^+^ T cells cannot detect and kill infected cells. Most therapeutic HIV vaccines rely on the CD8^+^ T cell response as the primary weapon to clear infected cells. Therefore, the latent reservoir is an unsolvable obstacle in the path of success for any therapeutic vaccine.

As it is now clear that a vaccine alone would not be enough to cure HIV, attention has turned into combining vaccines with other treatments, attacking the virus from multiple sides [91,92]. A strategy that has received much attention in recent years is the combination of therapeutic vaccines with latency reversal agents (LRA) and blocking antibodies. LRA are substances that reactivate the virus, making it visible for the attack of the immune system. Blocking antibodies, directed against the virus or the receptors used by the virus to enter the cells are used to limit the spread of the virus after latency is reversed. Based on that, our group is working on two different mRNA-based vaccines strategies.

Previous studies based on nucleic acids have reported the design of an HIV-1 T cell multi-epitope immunogen, termed DNA-TMEP-B. This immunogen codifies for eight HIV-1 fragments, mostly derived from conserved regions in Gag, Pol, and Nef proteins, restricted by a wide range of HLA class I and II molecules. These epitopes have been functionally associated with low viral load and HIV-1 control [6,7]. In these studies, preliminary data in mice showed that DNA plasmid encoding the polyepitope sequence (TMEP-B) was a potent inducer of the CD8+ T cell response. Our research group is testing the same sequence as mRNA (Project number: FIS PI18/00699) and in combination with immune modifying agents or LRAs as TLR7 agonist, IL15, and/or PD1/PDL1 pathway inhibitors [93,94].

Another example is the HIVACAR project (H2020 Grant Agreement Number: 731626) that will study a combined strategy on HIV-1 infected patients in a phase I/IIa clinical trial. The therapeutic approach will consist in (1) a personalized mRNA vaccine, (2) a blocking antibody against CD4, and (3) a LRA. (1) The studied vaccine will include a selection of the best epitopes of every participant, based on each patient’s HIV-1 reservoir and HLA-I alleles. (2) The 10-1074 is a monoclonal antibody that blocks the CD4 HIV-1 binding site in most of the HIV-1 strains described [95]. (3) Finally, romidepsin a histone deacetylase inhibitor, has been selected as the LRA [96]. Recruitment will start during the second half of 2021, and hopefully, we will be able to give better answers to this pandemic.

## 7. Conclusions

Despite all the improvements for mRNA-based vaccines that have been discussed before, it is now clear that if a functional cure of HIV is possible, it will need the combination of different interventions to be successful [91]. The latent viral reservoir is one of the major problems to cure HIV. The non-transcribing viruses that are integrated into the genome of cells are virtually invisible to the immune system. Therefore, even in the hypothetical case that the most optimized therapeutic vaccine is obtained, this will probably not be enough to eliminate the reservoir. Co-administration of a therapeutic mRNA vaccine with drugs that could reactivate the reservoir of HIV (Kick-and-Kill strategy), immune checkpoint inhibitors, or broadly neutralizing antibodies are all promising approaches [92]. Maybe the functional cure of HIV is in one of those combined treatments, but first, every single strategy will need to be optimized separately to make a combination successful.

## Figures and Tables

**Figure 1 viruses-13-00501-f001:**
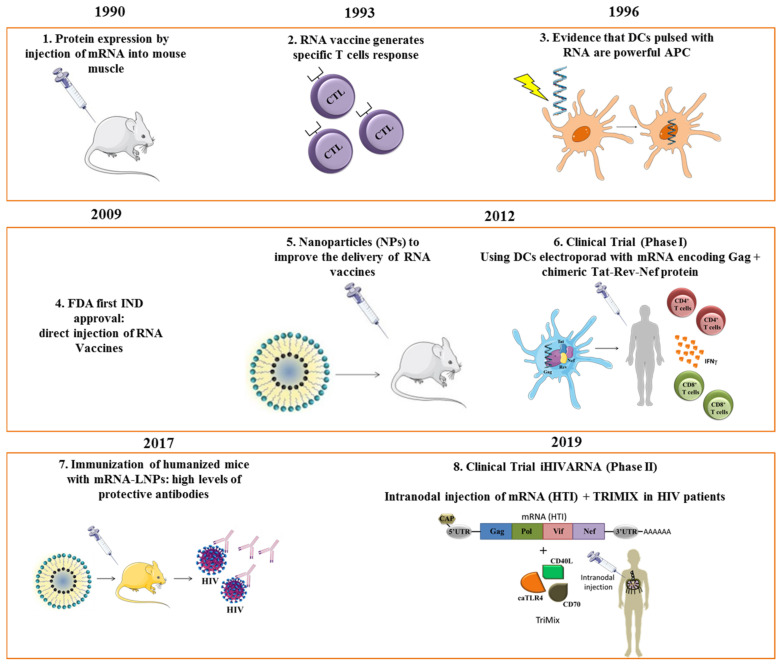
Milestones in RNA vaccines: steps and milestones in the development, application, and clinical use of mRNA-based vaccines. APC, antigen-presenting cell; DC, dendritic cells; NPs; nanoparticles; LNP, lipid nanoparticle; MVA, Modified Vaccinia Ankara. References for each time point: 1 [5], 2 [16], 3 [17], 4 [13], 5 [15], 6 [18], 7 [19], 8 [20].

**Figure 2 viruses-13-00501-f002:**
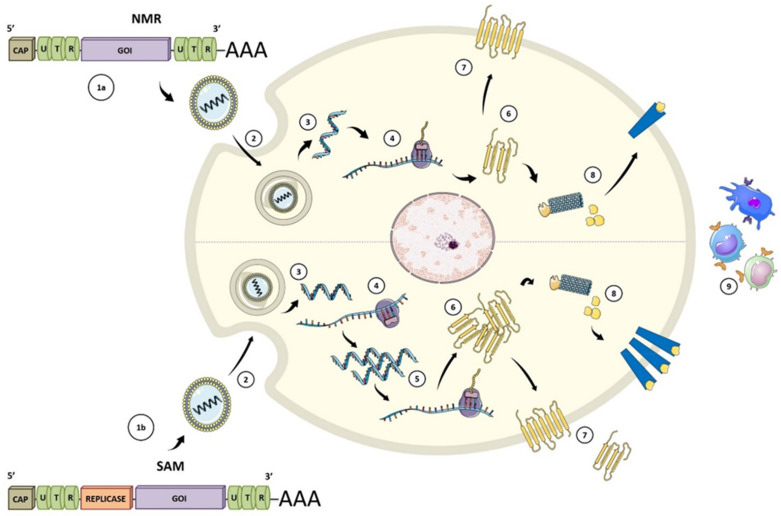
Types of mRNA vaccines. (1a) conventional or non-replicating (NRM) construct includes an opening frame encoding the gene of interest (GOI), 5′ and 3′ untranslated regions (UTRs), and a terminal poly(A). (1b) The self-amplifying mRNA (SAM) construct encodes replicative components to direct intracellular mRNA self-amplification and abundant protein expression. (2) Both structures required a delivery system, usually by endocytosis, for cellular uptake. Once the vaccine with its carrier is internalized (3), the mRNA is transported through the endosomal route and is released to the cytosol (4). NMR are immediately translated by ribosomes to produce the protein of interest. (5) SAM can also be translated by ribosomes to develop replicase machinery essential for self-amplification. (6) SAM mRNA constructs are translated to produce the protein of interest. (7) The expressed protein is generated in different ways: secreted, trans-membrane, or intracellular. (8) Protein processing for MHC presentation. (9) Peptide-MHC presentation and adaptive and innate immune responses after protein of interest detection. Figure adapted from [26,27].

**Table 1 viruses-13-00501-t001:** Advantages and disadvantages of RNA vaccines.

Vaccines	Advantages	Disadvantages
RNA	Non-infectious Non-integrating Cell free Rapid and scalable production	Instability

**Table 2 viruses-13-00501-t002:** Clinical trials of mRNA therapeutic vaccines against HIV.

Vaccine Description	Design	Main Findings
*AGS-004* Personalized immunotherapy using electroporation of DCs with autologous amplified HIV RNAs encoding Gag, Vpr, Rev, and Nef	(a) Pilot study—3 ID injections were administered every 4 weeks 4 times [14] (b) 2:1 randomized—4 ID injections every 4 weeks 4 times, underwent ATI and continue same vaccine schedule until treatment restart [76] (c) A sub-study (acute infection)—3 ID injections every month 5 times, underwent ATI and continued monthly dosing [77] (d) Pilot-study, combined with VOR as LRA—3 doses of VOR continued with 3 ID injections every 3 weeks over 12 weeks after 1 weeks followed with 10 consecutive doses of VOR at 72 h intervals [78]	− Safe, increased CD8 T cell proliferative response − Safe, no antiviral effect, robust expansion of CD28^+^/CD45RA CTL − Safe, all rebounded, increase CD28^+^/CD45RA CTL, an inverse correlate between TtR and proliferation − Safe, VOR had no effect on specific T cell response, vaccine effect on CTL was marginal
Autologous DCs electroporated with mRNA encoding Gag and a chimeric Tat, Rev, and Nef protein	Phase I/II study—ID and SC injections every 4 weeks 4 occasions [18]	− Safe, increase in magnitude and breadth of IFN-γ response to Gag, significant increase in proliferating T cells
Autologous DCs electroporated with mRNA encoding Gag and Nef	Randomized 2:1—4 ID injections at weeks 0, 2, 6, and 10, also received a contralateral ID injection of autologous DCs pulsed with KLH, a neo-antigen at weeks 0 and 2 [79]	− Safe, participants develop de novo CD4 and CD8 proliferative responses to KLH and CD4 proliferative responses to Nef
*iHIVARNA* naked mRNA-based vaccine encoding activation signals (TriMix: CD40L + CD70 + caTLR4) combined with rationally selected antigenic sequences [HIVACAT T-cell immunogen (HTI)] sequence comprises 16 joined fragments from Gag, Pol, Vif, and Nef)	(a) Dose escalation phase I clinical trial—3 intranodal injections ultrasound-guided every 2 weeks 3 times [81] (b) Phase IIa, randomized—3 intranodal injections ultrasound-guided every 2 weeks 3 times underwent 12 weeks ATI [80]	− Safe, induced moderate HIV-specific immune responses, transient increase in caHIV-RNA and usVL − Safe, no significant increase in the total frequencies of IFNγ+ for specific T cell responses in (these findings dictated to halt further inclusion for futility) The erroneous study product affects all conclusions

DCs: dendritic cells, ID: intradermal, ATI: antiretroviral treatment interruption, VL: viral load, CTL: cytolytic T lymphocytes, TtR: time to rebound, VOR: vorinostat, LRA: latency reverse agent, SC: subcutaneous, KLH: keyhole limpet hemocyanin, caHIV-RNA: cell-associated HIV-RNA, usVL; ultrasensitive viral load.

## Data Availability

Not applicable.

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
