# Peer review of "In the Era of mRNA Vaccines, Is There Any Hope for HIV Functional Cure?"

_viruses, 2021, doi:10.3390/v13030501_

Round 1

Reviewer 1 Report

This is a very interesting and appealing manuscript-review on mRNA-based vaccine platforms, with particular emphasis on the potential use of mRNA vaccines to prevent HIV rebound.   There are some comments that need to be addressed,  

  • Abstract sections reads “Some mRNA-based vaccines platforms against infection diseases have demonstrated encouraging results in mice and humans”. Considering that mRNA vaccines have also showed positive results in other animal models (i.e. monkeys) the authors should replace the work “mice” by animal models”. 
  • Figure 1 is very appealing but should contain at least one reference for each timeline within the figure
  • Figure 1 timeline 2008 reads “FDA approved”. For the general audience and to avoid confusion with licensing, the authors should re-write it as “FDA first IND approval”
  • Lines 76-99 discusses on DNA vaccines, but refers to Figure 1 that portraits history of RNA vaccines. This section and connect with Figure 1 needs to be clearer.
  • Lines 76-78 contains two duplicated sentences. Please erase one
  • Table 1. DNA vaccines are not cell-free since they are produced in bacteria. Please correct
  • Table 1 states that one of disadvantages of DNA vaccines is “poor immunogenicity” and one disadvantage for RNA vaccines is “low immunogenicity”. The authors should clarify the difference between “poor” and “low” when it comes to immunogenicity.
  • Line 140, spell pDNA and please keep consistency throughout the manuscript since in other sections reads as “plasmid DNA”
  • Lines 224-226 reads “Apart from presentation, the mRNA's product could also be secreted outside the cell as a source of antigen to produce antibodies”. Please correct since protein secreted by mRNA-transfected muscle cells that is taken-up by APCs, have been also proven to stimulate T cell responses.

  • Lines 227-233. TLR3 does not recognize single-strand RNA but rather double-strand RNA. The same for MDA-5 and RIG-1. The authors should specify if the statement relates to non-replicating RNA vaccines (single-strand) or replicating RNA (double strand) vaccines. Also RAGE is not known to recognize RNA. Please correct or alternatively add references to support the statement.
  • Lines 329-336. The paper that the authors are referring used CD34+ hematopoietic stem cells to generate in vitro dendritic cells (DCs) though the dendritic cells generated in vitro do not express CD34. As such the authors wording of CD34+DCs is incorrect. Please replace “CD34+DCs” by “CD34-derived DCs” or by “CD34-DCs”
  • Ref #35 is incomplete.

Author Response

Reviewer 1

This is a very interesting and appealing manuscript-review on mRNA-based vaccine platforms, with particular emphasis on the potential use of mRNA vaccines to prevent HIV rebound.  There are some comments that need to be addressed. Dear Reviewer 1, we appreciate your comments on our work. Down below there are our answers to your comments. The updated version of the document is attached as a "word document".

First, we want to note that the overall English of the manuscript has been improved.

1. Abstract sections reads “Some mRNA-based vaccines platforms against infection diseases have demonstrated encouraging results in mice and humans”. Considering that mRNA vaccines have also showed positive results in other animal models (i.e. monkeys) the authors should replace the work “mice” by animal models”.  As recommended the word “mice” has been replaced for “animal models”.

2. Figure 1 is very appealing but should contain at least one reference for each timeline within the figure. As recommended references for each timeline in figure 1 have been added.

3. Figure 1 timeline 2008 reads “FDA approved”. For the general audience and to avoid confusion with licensing, the authors should re-write it as “FDA first IND approval. As suggested “FDA approved” has been changed to “FDA first IND approval”.

4. Lines 76-99 discusses on DNA vaccines, but refers to Figure 1 that portraits history of RNA vaccines. This section and connect with Figure 1 need to be clearer. As recommended the reference for Figure 1 has been relocated to line 89, connecting the figure with RNA history. Also, the section of DNA vaccines has been simplified to make the introduction more focused on RNA vaccines.

5. Lines 76-78 contains two duplicated sentences. Please erase one. As suggested, this has been corrected.

6. Table 1. DNA vaccines are not cell-free since they are produced in bacteria. Please correct. As recommended this erroneous phrase has been eliminated. 

7. Table 1 states that one of the disadvantages of DNA vaccines is “poor immunogenicity” and one disadvantage for RNA vaccines is “low immunogenicity”. The authors should clarify the difference between “poor” and “low” when it comes to immunogenicity. As recommended, we have improved our information in table 1. We decided a different approach and just focused on mRNA vaccines advantages as compared with other vaccines not just DNA. This also solves the confusing description of immunogenicity.  

8. Line 140, spell pDNA and please keep consistency throughout the manuscript since in other sections reads as “plasmid DNA”. As suggested, we have corrected all inconsistencies through the manuscript. 

9. Lines 224-226 reads “Apart from presentation, the mRNA’s product could also be secreted outside the cell as a source of antigen to produce antibodies”. Please correct since protein secreted by mRNA-transfected muscle cells that is taken-up by APCs, have been also proven to stimulate T cell responses. As recommended, this phrase has been updated: The mRNA’s product could also be secreted outside the cell, being used as a source of antigen for antibody production and being internalized by APCs to be presented to TCD4s and cross-presented to TCD8s. Additional references to support this statement have been included (lines 155-158).

10. Lines 227-233. TLR3 does not recognize single-strand RNA but rather double-strand RNA. The same for MDA-5 and RIG-1. The authors should specify if the statement relates to non-replicating RNA vaccines (single strand) or replicating RNA (double strand) vaccines. As the reviewer suggests, this has been re-written, specifying which of the mentioned receptors bind to dsRNA and which receptors bind to ssRNA (lines 159-167). Also, RAGE is not known to recognize RNA. Please correct or alternatively add references to support the statement. As suggested, we have updated our references and included a manuscript by Bertheloot et al. that describes that RAGE could bind to RNA (ref#35).

11. Lines 329-336. The paper that the authors are referring used CD34+ hematopoietic stem cells to generate in vitro dendritic cells (DCs) though the dendritic cells generated in vitro do not express CD34. As such the authors wording of CD34+DCs is incorrect. Please replace “CD34+DCs” by “CD34-derived DCs” or by “CD34-DCs”. As suggested, we have replaced the term “CD34+DCs” with “CD34-DCs”. It has also been specified that CD34-DCs derive from CD34+ hematopoietic stem cells (lines 242-248).

12. Ref #35 is incomplete. The old reference #35 (now is #26), has been completed. 

Reviewer 2 Report

Esteban and colleagues in their review article described the hope and challenges of mRNA vaccine that could help in combating HIV infection. The review is important but needs major revision.

  1. The title should be changed. The authors claimed that it is a systemic review. However, there is no methodology presented in the review article that describes the inclusion and exclusion criteria of the articles used in the study. It seems to be a regular review article.
  2. Major portion of the review deals with mRNA vaccine per se. It should be dealt with mRNA vaccine in the HIV field. The writing should be tighter. Chapters 1 and 2 could be fused.
  3. The authors should describe the possible danger of introducing the mRNA vaccine. Possible recombination with the host genome and how to effective designing can prevent it.
  4. Figure 1 should describe the timeline of the anti-HIV mRNA vaccine rather than the whole timeline of mRNA vaccine development.
  5. A chapter describing the combinatorial approach to combat HIV infection. Can different types of vaccines (protein, mRNA) can have a better impact than mRNA only should be discussed?
  6. Table 1 and Table 2 should be redesigned. Too much text on a table does not serve the purpose of the table.
  7. HIV-1 latency is the major bottleneck in the cure of the infection. Is mRNA vaccine effective in such a situation should be discussed?

 Minor:

Functional not Fun-Tional in the title.

The review provides up-to-date information regarding the mRNA vaccine. However, authors should focus more on mRNA vaccine in the HIV area. The authors have done a great job and I recommended resubmission of the manuscript with addressing the above points. I will be happy to review the revised version.

Author Response

Reviewer 2

The review is important but needs major revision. The review provides up-to-date information regarding the mRNA vaccine. However, authors should focus more on mRNA vaccine in the HIV area. The authors have done a great job and I recommended resubmission of the manuscript with addressing the above points. I will be happy to review the revised version.

Esteban and colleagues in their review article described the hope and challenges of mRNA vaccine that could help in combating HIV infection.

Dear Reviewer 2, we appreciate your comments on our work. As suggested, we have made changes and focused our manuscript more on HIV, and also revised the overall English. The updated version of the manuscript is attached down below as a "word document".

1.The title should be changed. The authors claimed that it is a systemic review. However, there is no methodology presented in the review article that describes the inclusion and exclusion criteria of the articles used in the study. It seems to be a regular review article. As suggested, we have changed the title.

2. Major portion of the review deals with mRNA vaccine per se. It should be dealt with mRNA vaccine in the HIV field. The writing should be tighter. Chapters 1 and 2 could be fused. As suggested, we have tightened the writing.

3.The authors should describe the possible danger of introducing the mRNA vaccine. Possible recombination with the host genome and how an effective designing can prevent it. As suggested, we have reviewed the bibliography and we have found that most of the information states that the risk of integration for RNA is extremely low and using RNA for vaccination, in terms of integration, is thought as safe. However, we found a manuscript, not peer reviewed, that describes in vitro retro-transcription and integration of SARS-CoV-2 RNA (https://doi.org/10.1101/2020.12.12.422516). Taking into account all previous evidence and that the referred information has not yet been approved for publication or presented in any scientific meeting, we have not considered this information for the present manuscript.  

4.Figure 1 should describe the timeline of the anti-HIV mRNA vaccine rather than the whole timeline of mRNA vaccine development. As suggested, we have modified figure 1 and focused on HIV mRNA vaccines timelines.

5. A chapter describing the combinatorial approach to combat HIV infection. Can different types of vaccines (protein, mRNA) can have a better impact than mRNA only should be discussed? As suggested, we have included a chapter called “mRNA based functional cure strategies” in which the advantages of mRNA vaccine combined strategies versus vaccination alone are discussed.

6.Table 1 and Table 2 should be redesigned. Too much text on a table does not serve the purpose of the table. As recommended, we have redesigned the tables.

7.HIV-1 latency is the major bottleneck in the cure of the infection. Is mRNA vaccine effective in such a situation should be discussed? As recommended, we have included some discussion on HIV latency and its implication for a functional cure.

Minor:

8. Functional not Fun-Tional in the title. As suggested, this has been corrected.

Round 2

Reviewer 2 Report

The revised manuscript is much improved. I authors have incorporated the all changes suggested. The manuscript will be an important contribution in the area of HIV mRNA vaccine. The manuscript is well within the scope of the journal and may be accepted for publication. I congratulate the authors for the good job.